# Transcriptomic Analysis Reveals the Impact of the Biopesticide *Metarhizium anisopliae* on the Immune System of Major Workers in *Solenopsis invicta*

**DOI:** 10.3390/insects14080701

**Published:** 2023-08-11

**Authors:** Hongxin Wu, Yating Xu, Junaid Zafar, Surajit De Mandal, Liangjie Lin, Yongyue Lu, Fengliang Jin, Rui Pang, Xiaoxia Xu

**Affiliations:** National Key Laboratory of Green Pesticide, “Belt and Road” Technology Industry and Innovation Institute for Green and Biological Control of Agricultural Pests, College of Plant Protection, South China Agricultural University, Guangzhou 510642, China; scauwhx@stu.scau.edu.cn (H.W.); xuyt1101@stu.scau.edu.cn (Y.X.); jz_jaam@yahoo.com (J.Z.); surajit_micro@yahoo.co.in (S.D.M.); liangjie@stu.scau.edu.cn (L.L.); luyongyue@scau.edu.cn (Y.L.); jflbang@scau.edu.cn (F.J.)

**Keywords:** *Solenopsis invicta*, *Metarhizium anisopliae*, transcriptome, immune response

## Abstract

**Simple Summary:**

The red imported fire ant (*Solenopsis invicta*) is a highly invasive pest that causes significant damage to human health, agriculture, and biodiversity. Microbial insecticides have been identified as effective and promising alternatives for pest control. However, their efficacy is limited by the host’s immune system. In this study, we utilized RNA sequencing technology and bioinformatics methods to investigate gene expression changes in *S. invicta* infected with *Metarhizium anisopliae* at different time points (0, 6, 24, and 48 h). We identified 33 differentially expressed immune-related genes and constructed illustrative Toll and Imd signaling pathways diagrams. Our findings revealed that *M. anisopliae* suppresses the immune gene expression of *S. invicta* during the early stages of infection while inducing upregulation in the later stages. This suggests the potential of *M. anisopliae* as a potent biopesticide for controlling *S. invicta* populations. Overall, these findings provide a foundation for further understanding the immune mechanisms of *S. invicta* and its molecular response to *M. anisopliae*.

**Abstract:**

The red imported fire ant (*Solenopsis invicta* Buren, 1972) is a globally significant invasive species, causing extensive agricultural, human health, and biodiversity damage amounting to billions of dollars worldwide. The pathogenic fungus *Metarhizium anisopliae* (Metchnikoff) Sorokin (1883), widely distributed in natural environments, has been used to control *S. invicta* populations. However, the interaction between *M. anisopliae* and the immune system of the social insect *S. invicta* remains poorly understood. In this study, we employed RNA-seq to investigate the effects of *M. anisopliae* on the immune systems of *S. invicta* at different time points (0, 6, 24, and 48 h). A total of 1313 differentially expressed genes (DEGs) were identified and classified into 12 expression profiles using short time-series expression miner (STEM) for analysis. Weighted gene co-expression network analysis (WGCNA) was employed to partition all genes into 21 gene modules. Upon analyzing the statistically significant WGCNA model and conducting Kyoto Encyclopedia of Genes and Genomes (KEGG) pathway enrichment analysis on the modules, we identified key immune pathways, including the Toll and Imd signaling pathways, lysosomes, autophagy, and phagosomes, which may collectively contribute to *S. invicta* defense against *M. anisopliae* infection. Subsequently, we conducted a comprehensive scan of all differentially expressed genes and identified 33 immune-related genes, encompassing various aspects such as recognition, signal transduction, and effector gene expression. Furthermore, by integrating the significant gene modules derived from the WGCNA analysis, we constructed illustrative pathway diagrams depicting the Toll and Imd signaling pathways. Overall, our research findings demonstrated that *M. anisopliae* suppressed the immune response of *S. invicta* during the early stages while stimulating its immune response at later stages, making it a potential biopesticide for controlling *S. invicta* populations. These discoveries lay the foundation for further understanding the immune mechanisms of *S. invicta* and the molecular mechanisms underlying its response to *M. anisopliae*.

## 1. Introduction

The red imported fire ant (RIFA), *Solenopsis invicta* (Buren, 1972) (Hymenoptera: Formicidae), is considered one of the top 100 invasive pests worldwide [1] and has been shown to have negative impacts on human health, public safety, habitats, agriculture, and native biodiversity [2,3]. Since 2003, *S. invicta* has been invading China, causing significant losses to agricultural production in the southern region. The expansion rate of *S. invicta* in China reaches 26.5–48.1 km annually, and it has been reported that this invasive species has caused a total economic loss of USD 25 billion since its introduction to China [4,5]. *S. invicta* can affect agricultural production by damaging germinating seeds, seedlings, and growing crops [6]. Furthermore, *S. invicta* can also cause economic losses to agricultural production by displacing farmers and forcing them to abandon cultivated land [7]. In China, *S. invicta* is primarily found in habitats such as farmland and parks with frequent human disturbance [8]. In this scenario, the use of chemical insecticides to prevent or control *S. invicta* not only contaminates the environment but also poses risks to human health [9,10,11]. For example, chemical insecticides, such as organophosphates, can cause respiratory diseases, decreased lung function, and hematological changes through direct contact [12]. Therefore, there is a growing interest in environmentally friendly microbial insecticides, such as insect pathogenic fungi, for pest control [13,14].

Entomopathogenic fungi can attach to the insect’s cuticle, germinate, and penetrate the hemolymph, leading to fungal growth inside the host, ultimately producing and dispersing secondary infectious conidia [15]. Among these *Beauveria bassiana* (Bals.-Criv.) vuill. (1912) [16] and *Metarhizium anisopliae* (Metchnikoff) Sorokin (1883) have shown potential against various pests, including *Aedes albopictus* (Skuse, 1894) [17], *Bemisia tabaci* (Gennadius, 1889) [18], *Spodoptera frugiperda* (J. E. Smith, 1797) [19], *Locusta migratoria* (Linnaeus, 1758) [20], *Alphitobius diaperinus* (Panzer, 1797) [21], *Nilaparvata lugens* (Stål, 1854) [22], and *Plutella xylostella* (Linnaeus, 1758) [23]. However, the effectiveness of these biopesticides is often limited by the defense systems of insect pests [24]. For example, the immune system of *S. invicta* can effectively respond to infection by SINV-3 (*S. invicta* virus-3) [25]; *Monochamus alternatus* (Hope, 1842) can activate the Toll pathway to resist invasion when infected by pathogenic fungi [26]. To develop effective biopesticides, a deeper understanding of the molecular mechanisms of the pest defense system is essential. Insects respond to pathogenic infection by initiating various immune pathways, activating transcription factors, and altering gene expression [27]. In contrast to the mammalian system, they rely solely on innate immunity, divided into cellular and humoral immune responses. The cellular innate immune response is mediated by the robust phagocytic activities of plasmacytes, whereas melanin synthesis, clotting, and the production of antimicrobial peptides, orchestrated by the fat body, collectively constitute humoral innate immunity [28].

Although much work has been performed on insect-pathogen interaction, little is known about the immune defense mechanism of *S. invicta* [21,29,30]. The present study aims to elucidate the immune responses of *S. invicta* during *M. anisopliae* by high-throughput RNA-seq analysis. Our analysis suggests the existence of an intricate regulatory interplay between *S. invicta’s* immune system and *M. anisopliae*, wherein the latter might modulate *S. invicta’s* immune response via toxins to promote its survival. By exploiting these characteristics, we can explore the possibilities of enhancing the potential of existing biopesticides, offering a novel approach to the development of biopesticides. 

## 2. Materials and Methods

### 2.1. Insects

Colonies of *S. invicta* were collected from Huangpu District (Guangzhou, Guangdong, China) and established in the laboratory. A plastic container (20 L) coated with Fluon^®^ on its inner wall was employed as the nest arena. Separate Petri dishes containing a cotton ball saturated with honey-water solution (25%) were placed in the container. *Tenebrio molitor* (Linnaeus, 1758) was provided as a diet. Colonies were maintained at 26 °C, 12 h light:12 h dark photoperiod, and 70% relative humidity.

The division of labor among *S. invicta* is related to the age and size of workers. Major workers, the largest in the colony with powerful jaws, are responsible for defense. Worker ants with a head width > 1.000 mm were considered major workers [31]. Head width was measured using a digital micrometer (FB70252; Thermo Fisher Scientific, Waltham, MA, USA) under a stereomicroscope (SMZ-1500; Nikon, Tokyo, Japan) [32].

### 2.2. M. anisopliae Infection in S. invicta

The entomopathogenic fungus, *M. anisopliae*, MaqS1902, was kindly provided by Qiongbo Hu (South China Agricultural University, Guangzhou, China). Monoconidial culture (10 days) grown on potato dextrose agar (PDA) was harvested with a disinfected spatula in 0.05% (*v*/*v*) Tween-80 (CMC = 0.012 mM (Sigma Aldrich P1754)) and diluted in sterilized distilled water. The spores were calculated using a hemocytometer; stock solutions were prepared and stored at 4 °C until the desired concentrations were made [33].

To ensure the quality of bioassays, we placed 180 major worker ants in disinfected plastic dishes (9 cm diameter). After 2 h, we carefully observed the worker ants, removed any individuals that had died from mechanical damage, and ensured each group contained 150 active and healthy major worker ants.

Five concentrations (1.00 × 10^8^, 1.00 × 10^7^, 1.00 × 10^6^, 1.00 × 10^5^, 1.00 × 10^4^ spores/mL) were prepared (hit and trial method) while 0.05% Tween-80 was used as the control. *M. anisopliae* was applied by dipping the major worker ants into the desired concentrations for 2–3 s. After dipping, the ants were transferred to filter paper for drying and placed in disinfected plastic dishes (9 cm diameter). 150 major worker ants were exposed to each concentration, and a sufficient diet was provided throughout the experimentation. Adults without movements were considered deceased. The mortality data were recorded over 10 days, adjusted using Abbott’s formula, and compared using the log-rank test in SPSS30 to assess differences in mortality rates. This experiment was repeated four times.

*S. invicta* individuals were infected with *M. anisopliae* using the aforementioned infection method at a concentration of 5.00 × 10^7^ spores/mL (LC_50_). Randomly selected *S. invicta* individuals were observed under a stereomicroscope (SMZ-1500; Nikon, Tokyo, Japan) at 0, 6, 24, 48, and 72 h post-infection. Subsequently, deceased *S. invicta* specimens were placed in a humid incubator for 96 h to monitor spore development. At the same time, *S. invicta,* which died naturally, was also incubated under the same conditions as a control.

### 2.3. RNA Extraction and Transcriptome Sequencing

Based on our preliminary experiments, a concentration range of 10^4^–10^10^ of *M. anisopliae* could induce mortality. To explore the impact of *M. anisopliae* on the immune system of major workers of *S. invicta*, we conducted Real-time quantitative PCR (RT-qPCR) experiments. Notably, we observed significant changes in the expression of key factors, such as AMPs and PPO, when treated with 5.00 × 10^7^ spores/mL of *M. anisopliae* at 48 h. Consequently, we decided to focus our sampling on *M. anisopliae*-infected major workers treated with 5.00 × 10^7^ spores/mL to investigate the immune mechanisms of *S. invicta*.

The samples were collected at 0, 6, 24, and 48 h post-infection, with three replicates at each time point. In total, 12 libraries were constructed. The libraries were constructed using the following method. Total RNA was extracted using a Trizol reagent kit (Invitrogen, Carlsbad, CA, USA) according to the manufacturer’s protocol. RNA quality was assessed on an Agilent 2100 Bioanalyzer (Agilent Technologies, Palo Alto, CA, USA) and checked using RNAase-free agarose gel electrophoresis. After the total RNA was extracted, rRNAs were removed, and the enriched mRNAs were fragmented into short fragments using a fragmentation buffer and reverse transcribed into cDNA with random primers. Second-strand cDNA was synthesized using DNA polymerase I, RNase H, dNTPs (dUTP instead of dTTP), and buffer. Next, the cDNA fragments were purified with a QiaQuick PCR extraction kit (Qiagen, Venlo, The Netherlands), end-repaired, poly(A) added, and ligated to Illumina sequencing adapters. Then UNG (Uracil-N-Glycosylase) was used to digest the second-strand cDNA. The digested products were size selected by agarose gel electrophoresis, PCR amplified, and sequenced using Illumina NovaSeq 6000 by Novogene Biotechnology Co. (Beijing, China).

The raw reads in fastq format were filtered using fastp [34] (version 0.18.0) to obtain clean data. Quality statistics, including Q20, Q30, and GC content, were calculated, and the resulting clean reads were mapped to the reference genome of *S. invicta* (https://www.ncbi.nlm.nih.gov/data-hub/genome/GCF_016802725.1/) (accessed on 3 April 2023) [35] using HISAT2 (v2.1.0) [36]. Gffcompare (version 0.12.6) [37] was used to count the number of reads mapped to each gene, and then FPKM was calculated. To annotate the obtained genes and predict their functions, a comparative analysis was performed using the Blast program against the NR database. This approach allowed for the identification of homologous sequences and provided valuable information for the functional prediction of the genes under investigation. To assess the reproducibility of each sample, Pearson’s correlation analysis and principal component analysis were performed using the OmicShare online data analysis platform (https://www.omicshare.com/tools) (accessed on 17 April 2023).

### 2.4. Differential Gene Expression Analysis

The differential expression analysis of RNAs between five different groups was conducted using the software DESeq2 [38], whereby genes with an FDR < 0.05 and |log2FC| > 1 were considered as differentially expressed genes (DEGs). Kyoto Encyclopedia of Genes and Genomes (KEGG) pathway enrichment analysis for DEGs was performed using the ClusterProfiler package [39]. To identify immune-related genes among the DEGs, we searched for immune-related genes from the DEGs based on gene annotation and KEGG enrichment results. Genes annotated as hypothetical or unknown proteins were excluded.

### 2.5. Temporal Analysis

The short time-series expression miner software (STEM v1.3.13) [40] was used for analyzing the expression trends of DEGs. The pre-processing of data was performed using the “Log normalize data” option, with profiles set to 12. Profiles having *p* < 0.05 were defined as significantly trending. KEGG pathway enrichment analysis was performed on the gene set within the significantly trending profiles.

### 2.6. Weighted Co-Expression Network Analysis

A weighted gene co-expression network analysis was performed using the R package WGCNA (v1.47) [41] to identify gene modules that exhibited strong correlations with the hours post-infection (HPI) variable, followed by KEGG pathway enrichment analysis. Prior to importing the WGCNA, gene expression values were filtered (FPKM < 0.3, in half or more samples) using default settings, except for power being set at 11, TOM Type at unsigned, merge cut height at 0.25, and min module size at 50. From the relevant module, genes associated significantly with HPI (*p* < 0.05) and involved in the Toll and Imd signaling pathways were selected. These genes were further represented using pathway diagrams.

### 2.7. RT-qPCR Analysis

To validate the results of RNA-Sequencing, RT-qPCR was employed. Eight DEGs (*GNBP1*, *GNBP3*, *PGRP-SC2*, *ModSP1*, *Toll3*, *IMD*, *Relish*, and *Cactus*) from the Toll and Imd signaling pathway were selected. The total RNA was isolated from the major worker ants, as described earlier. First-strand cDNA (1 μg) was prepared using M-MLV reverse transcriptase (Promega, Madison, WI, USA), following the instruction manual. RT-qPCR was carried out on a Bio-Rad iQ2 optical system (Bio-Rad, Hercules, CA, USA) using the SsoFast EvaGreen Supermix (Bio-Rad, USA), following the instruction manual. The *RPL18* gene was used as an internal control for normalization. The purity of the PCR products was confirmed by generating a dissociation curve from 65 °C to 95 °C with the following PCR conditions: 95 °C for 30 s, 40 cycles at 95 °C for 5 s, and 55 °C for 10 s. The primers used in this study are listed in Appendix A. Each treatment included three replicates, and each reaction was run in triplicate. Data analysis was performed using the 2^−ΔΔCT^ method [42].

## 3. Results

### 3.1. M. anisopliae Infection in S. invicta

In the current study, we exposed *S. invicta* to different concentrations of *M. anisopliae* and recorded mortality over 10 days. Highly concentration-dependent results were observed (Figure 1A). At a concentration of 1.00 × 10^8^ (spores/mL), the final mortality rate of *S. invicta* after infection was 91.33%, while it was 62.00% at 1.00 × 10^7^ (spores/mL) and 41.27% at 1.00 × 10^6^ (spores/mL). However, no significant difference was observed in mortality rates between the control group and the 1.00 × 10^4^ (spores/mL) concentration. The 4-day LC_50_ was determined as 4.93 × 10^7^ (spores/mL), and the 5-day LC_50_ was 3.74 × 10^7^ (spores/mL). For subsequent RNA-seq experiments, a concentration of 5.00 × 10^7^ spores/mL was used.

To confirm fungal pathogenicity, we closely monitored the mycelial growth on the external surface of *S. invicta* after *M. anisopliae* infection from 24 to 72 h (Figure 1B). Mycelial growth could be observed on the head, thorax, abdomen, and abdominal leg joints as time progressed. Once dead, the carcasses were placed in a humid chamber to monitor additional conidial growth. Within 96 h (Figure 1C), we observed dense spore formation of *M. anisopliae* on the carcasses of *S. invicta*, further confirming the pathogenic effect.

### 3.2. Overview of RNA-Seq Data

The transcriptomic sequencing performed on 12 samples of *S. invicta* generated 141.50 GB of raw data. The samples included major worker ants from the control (0.05% Tween-80) and major worker ants from *M. anisopliae*-infected group at different time intervals (6 h, 24 h, and 48 h). After raw data filtration, we obtained 140.55 GB of high-quality data. Each sample had a size greater than 11.95 GB, with GC content ranging from 41.79% to 45.34% and a Q30 base percentage between 91.58% and 94.00% (Appendix A).

Subsequently, we constructed an expression profile containing 14,941 genes from the sequencing results of 12 samples. Pearson correlation analysis was performed, and the results for each biological replicate ranged from 0.92 to 0.99, indicating robust reproducibility of samples (Figure 2A). The conclusion drawn from the principal component analysis (PCA) suggests that the three replicates can be grouped, whereas there exist considerable variations when comparing the control group (0 h) with the treatment groups (6 h, 24 h, and 48 h) (Figure 2B).

### 3.3. Differential Gene Expression Analysis

To gain a better understanding of the gene expression in *S. invicta* during *M. anisopliae* infection, five groups of DEGs were identified at different stages of infection, including 0 h vs. 6 h, 0 h vs. 24 h, 0 h vs. 48 h, 6 h vs. 24 h, and 24 h vs. 48 h, with a total of 1313 DEGs. The total number of DEGs was 619 at 0 h vs. 6 h (311 upregulated and 308 downregulated), 650 at 0 h vs. 24 h (323 upregulated and 327 downregulated), 841 at 0 h vs. 48 h (396 upregulated and 445 downregulated), 25 at 6 h vs. 24 h (10 upregulated and 15 downregulated), and 274 at 24 h vs. 48 h (115 upregulated and 159 downregulated) (Figure 3A). In addition, by mapping all DEGs to KEGG reference pathways, their potential involvement in biological pathways was determined (Figure 3B). In all comparisons between the control and treatment groups (0 h vs. 6 h, 0 h vs. 24 h, and 0 h vs. 48 h), multiple KEGG pathways related to lipid synthesis and metabolism were significantly enriched. These pathways included terpenoid biosynthesis, fatty acid metabolism, biosynthesis of unsaturated fatty acids, and fatty acid biosynthesis. In the comparison between 24 h and 48 h, the lysosome pathway and Toll and Imd signaling pathways were enriched.

### 3.4. Temporal Expression Trends during the Infection

Short time-series expression miner (STEM) analysis was employed to examine the expression patterns of DEGs, and statistically significant profiles were visualized using colors. Five statistically significant profiles were identified (Figure 4A), and these profiles displayed a regular fluctuation in gene expression during *M. anisopliae* infection. KEGG enrichment analysis was performed on the five significant profiles, and the results revealed that downregulated profiles (Profiles 0 and 1) were enriched in pathways related to metabolism, while immune-related pathways were identified in Profiles 1, 2, and 11. Profiles 0 and 1 are mainly involved in synthesizing and metabolizing various biological molecules such as arginine and proline metabolism, arachidonic acid metabolism, steroid biosynthesis, and fatty acid metabolism. Additionally, Profile 1 also significantly enriched the lysosome pathway. Profiles 2 and 11 significantly enriched the Toll and Imd signaling pathways (Figure 4B).

### 3.5. Co-Expression Network Associated with S. invicta Infected by M. anisopliae

To gain a deeper understanding of the underlying gene expressional changes involved in *S. invicta’s* defense against *M. anisopliae* infection, we performed a WGCNA analysis. After filtering out low-expression genes (FPKM < 0.3 in half or more samples), a total of 11,872 genes were analyzed. Here, we identified 21 modules and calculated the correlation coefficient between module eigengenes and infection time (Figure 5A). Five modules significantly correlated with HPI (*p* < 0.05): orange3, dark-orange, sienna3, black, and ivory. Among them, the orange3, dark-orange, and sienna3 modules positively correlated with HPI, and the black and ivory modules negatively correlated. KEGG pathway enrichment analysis was also performed on the significantly correlated modules (Figure 5C). The negatively correlated modules (black and ivory) were mainly enriched in metabolism (blue class) and genetic information processing (green class). In contrast, the positively correlated modules (orange3, dark-orange, and ivory) were primarily enriched in metabolism (blue class), organismal systems (olive class), environmental information processing (purple class), and cellular processes (pink class). The dark-orange module enriched several immune-related pathways, including the Toll and Imd signaling pathways, lysosomes, autophagy, and phagosomes.

### 3.6. Identification of Immunity-Related Genes

A total of 33 immune-related genes were identified from the 1313 DEGs and classified into five categories: recognition, signal transduction, melanization, effectors, and others (Appendix A). Amongst these 33 immune-related genes, 1, 2, and 10 were upregulated, while 8, 8, and 9 were downregulated at 6, 24, and 48 h post-treatment, respectively (Figure 6A). We constructed a heatmap of the average expression levels of these genes (Figure 6B). The heatmap showed that most differentially expressed immune genes in *S. invicta* were downregulated at 6 and 24 h after *M. anisopliae* infection. However, at 48 h post-infection, significant upregulation was observed in some genes. Based on the results of the STEM analysis, the majority of these immune-related genes were categorized under Profiles 2 and 10, elucidating the expression patterns of immune genes in *S. invicta*. These expression patterns unveiled that, apart from the upregulation upon infection by *M. anisopliae*, there was also an observed trend of initial downregulation followed by subsequent upregulation. Additionally, integrating the results from WGCNA analysis revealed that genes involved in pathogen recognition and signal transduction mainly belonged to the dark-orange module (positively correlated with HPI). In contrast, among the genes involved in modulation, most serine proteases are classified into the black module (negatively correlated with HPI), while their corresponding inhibitors, serpins, were predominantly present in the dark-orange module (Appendix A).

### 3.7. Expression of Genes Related to Toll and Imd Signaling Pathway

The Toll and Imd signaling pathways are crucial in insect humoral immunity and play a significant role in combating invading pathogens. Therefore, a detailed pathway analysis was conducted on the components of these pathways. Upon integrating the significantly correlated module genes identified in the above WGCNA analysis with the DEGs, we constructed an expression network diagram of the Toll and Imd signaling pathways after *M. anisopliae* infection in *S. invicta* (Figure 7).

In the WGCNA analysis, *GNBP1*, *PGRP-SA*, *ModSP1*, *ModSP2*, *Spaetzle1*, *Spaetzle2*, *Toll3*, *Cactus*, *PGRP-LC*, *IMD*, *Ird5*, and *Ankyrin* were assigned to the dark-orange module, indicating a significant positive correlation between their expression patterns and HPI. On the other hand, *Myd88*, *Pelle*, *Tube*, *Dorsal*, and *Defensin2* were assigned to the dark module in the WGCNA analysis, indicating a significant negative correlation between their expression patterns and HPI. Meanwhile, *GNBP3*, *Relish*, *PGRP-SC2*, *Hymenoptaecin1*, and *Hymenoptaecin2* only belonged to the DEGs. The gene information annotated in the pathway diagrams can be found in Appendix A. To assess the reliability of the transcriptome data, eight DEGs (*GNBP1*, *GNBP3*, *PGRP-SC2*, *Relish*, *ModSP1*, *Toll3*, *IMD*, and *Cactus*) were randomly selected for RT-qPCR analysis. The results demonstrated that the expression trends of all selected genes were consistent with RNA-Seq data (Appendix A).

## 4. Discussion

In the present study, we observed that *M. anisopliae* could effectively kill *S. invicta*, and produce secondary infectious spores, indicating its potential as a potent biocontrol agent for managing *S. invicta* infestations. Our research also revealed a dual effect on the immune response of *S. invicta* during the infection process. While *M. anisopliae* suppressed the immune response during the early stages of infection, it stimulated the immune response at later stages. These findings highlight the importance of considering the host immune responses when developing effective and sustainable pest management strategies.

The widespread use of chemical insecticides raises the risk of resistance development in *S. invicta*, contributes to environmental pollution, and negatively affects non-target organisms [43,44]. To reduce the reliance on synthetic chemicals, the application of pathogenic microorganisms has gained considerable attention. The soil-borne pathogenic fungus *M. anisopliae* is widely adopted for pest control due to its eco-friendly and non-toxic characteristics [45]. As a broad-spectrum fungal insecticide, *M. anisopliae* has been extensively used to control various social insect pests, including termites, ants, and wasps [46,47,48,49]. There have been studies on the use of *M. anisopliae* for controlling *S. invicta* [50]. To defend against invading pathogens, *S. invicta* has developed a series of behavioral and physical adaptations at colony levels, led by major worker ants, the strong and muscular defenders of the colony. However, apart from social defense, colony individuals also have innate immune responses, which have received little attention. Here, we try to decipher the underlying molecular interaction between the entomopathogenic fungus, *M. anisopliae*, and the major worker of *S. invicta* [51]. To investigate, we performed transcriptome sequencing on *S. invicta* samples at different time points post-infection. A total of 1313 DEGs were identified, and KEGG enrichment analysis was conducted on these DEGs. The analysis revealed enrichment in several immune-related pathways, including lysosomes and the Toll and Imd signaling pathways, indicating that *M. anisopliae* infection can influence the immune response of *S. invicta*. The lysosome is associated with processes such as phagocytosis, and autophagy, which aim to eliminate invading pathogens [52]; Toll and Imd signaling pathways detect invading pathogens, activate the humoral immune response, and produce AMPs to combat pathogens [53].

STEM software and the WGCNA method have found widespread application in entomology for transcriptome data analysis, revealing insights and patterns within insect transcriptomes and contributing to our understanding of gene expression dynamics [54,55,56]. The STEM analysis (Figure 4) identified immune-related pathways in Profiles 1, 2, and 11. These included Toll and Imd signaling pathways, lysosomes, and arachidonic acid metabolism. The Toll and Imd signaling pathways were significantly enriched in Profiles 2 and 11, which play crucial roles in the insect humoral immune response. The genes involved in the Toll and Imd signaling pathways exhibited sustained upregulation after infection to combat pathogens (Profile 11) but also showed transient downregulation during the early stages of infection, followed by a return to normal expression levels (Profile 2). This suggests that the humoral immunity of *S. invicta* may be partially suppressed following *M. anisopliae* infection. The lysosome, significantly enriched in Profile 1, exhibited lower expression levels than control after infection. Arachidonic acid can be converted into prostaglandins, and other eicosanoids, which mediate insect immune responses [57]. It has been reported that dietary arachidonic acid supplementation increases the expression of immune genes in *Apis mellifera* (Linnaeus, 1758) [58]. Arachidonic acid metabolism, significantly enriched in Profile 1, represents a group of genes downregulated early after infection. We hypothesize that the downregulation of arachidonic acid metabolism may indicate a weakened immune response of *S. invicta* to fungal infection. STEM analysis revealed the expression trends of DEGs, while WGCNA analysis encompassed genes beyond DEGs, enabling us to uncover additional potential pathways involved in *S. invicta* defense against *M. anisopliae* infection (Figure 5). The WGCNA analysis revealed that the genes included in the model that negatively correlated with HPI were associated with metabolism and genetic information processing, suggesting that *M. anisopliae* infection restricts these functional aspects in *S. invicta*. On the contrary, the genes included in the model that positively correlated with HPI were predominantly related to organismal systems, environmental information processing, and cellular processes, indicating their potential induction by *M. anisopliae* infection. Interestingly, the dark-orange module was enriched with many immune-related pathways, including the Toll and Imd signaling pathways, lysosomes, autophagy, and phagosomes. These pathways have been reported as relevant to insect immune responses [59,60,61]. It is important to note that the genes involved in these pathways are assigned to the same model, and genes assigned to the same model exhibit similar expression patterns. This suggests a connection between these pathways, indicating their collective involvement in resisting *M. anisopliae* infection.

The above analysis (STEM and WGCNA) provides a comprehensive perspective on the immune response of *S. invicta* against *M. anisopliae* from the pathway point of view. Subsequently, we investigated the crucial immune-related genes involved in the defense system of *S. invicta* against *M. anisopliae* infection. A total of 33 immune-related genes (Appendix A) were identified from the 1313 DEGs and were categorized into recognition, signal transduction, modulation, effectors, and others (Figure 6B). Compared to other insects, *S. invicta* has fewer immune-related genes, a shared characteristic in social insects [62]. Interestingly, among these immune-related genes, in addition to being continuously upregulated upon *M. anisopliae* infection, some genes showed a trend of initial downregulation followed by upregulation. It has been previously reported that secondary metabolites produced by *M. anisopliae*, such as destruxins, can inhibit insect immune responses [63]. A similar upregulation of immune genes has been observed at a specific time point (36 h after infection) during *B. bassiana* infection in *P. xylostella*, which is considered a critical time window for infection [64]. The immune recognition families play a crucial role in defending against invading microorganisms and triggering cellular and humoral responses. Pathogens encode conserved pathogen-associated molecular patterns (PAMPs), while hosts produce pattern recognition receptors (PRRs) as a response mechanism [65]. PRRs such as peptidoglycan recognition proteins (*PGRPs*), β-glucan binding proteins (*GNBPs*), and scavenger receptors can bind to PAMPs [66], triggering downstream immune reactions. For instance, *PGRPs* and *GNBPs* can activate the production of AMPs through the Toll and Imd signaling pathways in insects [67]. The present study identified three *PGRPs*, two *GNBPs*, and one scavenger receptor from the DEGs in *S. invicta*. *GNBP1* and *PGRP-SA1* exhibited initial downregulation followed by upregulation, while *PGRP-SA2* and *SCARB1* showed exclusive upregulation in the later stages, and *GNBP3* and *PGRP-SC* displayed an overall decreasing trend after *M. anisopliae* infection. Overall, the expression trends of the recognition families demonstrated early suppression followed by promotion later. Once the PAMPs of *M. anisopliae* are recognized by the PRRs of *S. invicta*, innate immune responses, such as phagocytosis and melanization, are immediately initiated. At the same time, multiple intracellular signaling pathways, including Toll and Imd signaling pathways, generate active substances such as AMPs and reactive oxygen species to eliminate invading pathogens. In the early stages, downregulation of PPRs indicates suppression of the innate immunity in *S. invicta*, which facilitates colonization of *S. invicta* by *M. anisopliae*. However, these PPRs begin to express in the later stages of infection. Our results suggest that the immune response of *S. invicta* to *M. anisopliae* is highly complex and represents a trade-off between the two species. We hypothesize that initially, *M. anisopliae* releases secondary toxins, such as destruxin A, which suppress the immune response to successfully infect *S. invicta* [68]. At the same time, numerous other pathogens can also infect *S. invicta*, and all these pathogens compete for resources. In such cases, *M. anisopliae* may reduce the release of destruxin A, triggering the innate immune response of *S.invicta* to resist other pathogens, thereby allowing *M. anisopliae* to survive for an extended period.

Melanization, another vital component of the innate immune system, is regulated by the phenoloxidase (*PO*) cascade reaction mediated by prophenoloxidase (*PPO*) [69]. Upon pathogen invasion, *PPO* is activated by the PPO-activating factor (*PPAF*), leading to the conversion of *PPO* into *PO*. Subsequently, quinone intermediates are generated, effectively eliminating the pathogens. We identified two key players in the present study, *PPAF* and *PPO*. Intriguingly, both were significantly downregulated in the early stages after infection (6 and 24 h). Serpins, a crucial superfamily of protease inhibitors, play a fundamental role in various physiological functions [70]. They act as negative regulators of the Toll pathway and melanization cascade by inhibiting serine proteases, thus preventing excessive immune damage [71,72]. Interestingly, our analysis identified seven differentially expressed serine proteases and four serpins. Most of the serpin genes exhibited upregulation during the infection process, while the majority of serine proteases displayed downregulated expression. Collectively, our research findings indicate that the infection of *M. anisopliae* may inhibit the cascade reaction of melanin synthesis in *S. invicta* by inducing the expression of serine protease inhibitors and suppressing the expression of serine proteases, which could be a potential reason for suppressing of phenoloxidase cascade reaction. Similar observations have also been reported in the case of *P. xylostella* following exposure to the *M. anisopliae* toxin destruxin A [73].

In *Drosophila melanogaster* (Meigen, 1830), the regulation of antifungal peptides in response to fungal infection primarily relies on the Toll pathway; mutations in the Toll pathway render *D. melanogaster* more susceptible to fungal infections and impair the inducible production of the antifungal peptide, drosomycin [74]. Nevertheless, in *A. mellifera*, it is widely acknowledged that the Toll pathway controls their antifungal response. In addition to Toll, simultaneous activation of the Imd, JNK, and JAK/STAT pathways occurs upon fungal infection, resulting in a remarkable surge in the production of specific AMPs, including *abaecin*, *defensin-2*, and *hymenoptaecin* [75]. In the present study, we also observed that the expression of the heterotrimeric death domain complex (*Myd88*, *Pelle*, and *Tube*) was consistently lower after infection compared to the pre-infection levels. The heterotrimeric death domain complex plays a crucial role in the signal transduction of the Toll pathway, contributing to the antifungal innate immune response in insects [76,77]. Furthermore, the expression of the negative regulatory factor *Cactus* was significantly upregulated after 48 h infection. Cactus protein acts as an inhibitor and regulates the production of AMPs by binding to the NF-κB family protein Dorsal, preventing its entry into the nucleus. The elevated expression of *Cactus* maintains the expression levels of corresponding AMPs. Previous studies have reported that the upregulation of Cactus protein expression in the fat body of *Aedes aegypti* (Linnaeus, 1762) after *M. anisopliae* infection, which helps maintain the expression levels of AMPs, such as *Defensin* A and *Cecropins* G, prior to the infection [78]. Correspondingly, the Toll pathway-regulated AMP, *Defensin*, exhibited lower expression levels during infection compared to pre-infection levels. It indicates that the ability of *S. invicta* to activate the Toll signaling pathway is inhibited upon *M. anisopliae* infection; this inhibition might favor the successful infection of *S. invicta* by *M. anisopliae*. In the future, we can use genetic engineering techniques to generate recombinant *M. anisopliae* strains that silence key genes in the Toll signaling pathway, thereby obtaining highly efficient and environmentally friendly biopesticides.

The *IMD* and *Ird5* kinase play important roles in the Imd pathway, with the protein encoded by the *IMD* gene participating in signal transduction and *Ird5* kinase serving as a key enzyme in the Imd pathway [79,80]. We observed that the *IMD* and *Ird5* kinase expression was significantly upregulated at 48 h. The NF-κB family transcription factor *Relish* was initially downregulated after infection but recovered to pre-infection levels. The expression of Ankyrin, an anchor protein that positively regulates the Imd signaling pathway, showed a gradual increase. The downregulation of the NF-κB family transcription factor *Relish* after infection may be a mechanism to suppress the immune response. Furthermore, the expression of *Hymenoptaecin* initially decreased post-infection but later recovered to pre-infection levels. In Hymenoptera, *Hymenoptaecin* is believed to be regulated by the Imd pathway [81].

## 5. Conclusions

Our research findings demonstrated that *M. anisopliae* suppressed the immune response of *S. invicta* during the early stages (6 and 24 h) while stimulating its immune response at later stages (48 h). In subsequent studies, it is imperative to confirm the function of these immune-related genes by RNAi, miRNA, or lncRNA. There remain challenges concerning the field efficacy of employing *M. anisopliae* for controlling *S. invicta*. For instance, weather conditions can affect the germination of *M. anisopliae* spores, and *S. invicta* actively engages in self-grooming and maintains nest hygiene. In order to address these issues, the ability of *M. anisopliae* to suppress the *S. invicta’s* immune defense mechanisms during early infection stages can be utilized by incorporating it with existing microbial insecticides (such as viruses or *Bt*). Another viable approach is to construct recombinant fungi through genetic engineering, which would result in lethal and fast-acting bioinsecticides.

## Figures and Tables

**Figure 1 insects-14-00701-f001:**
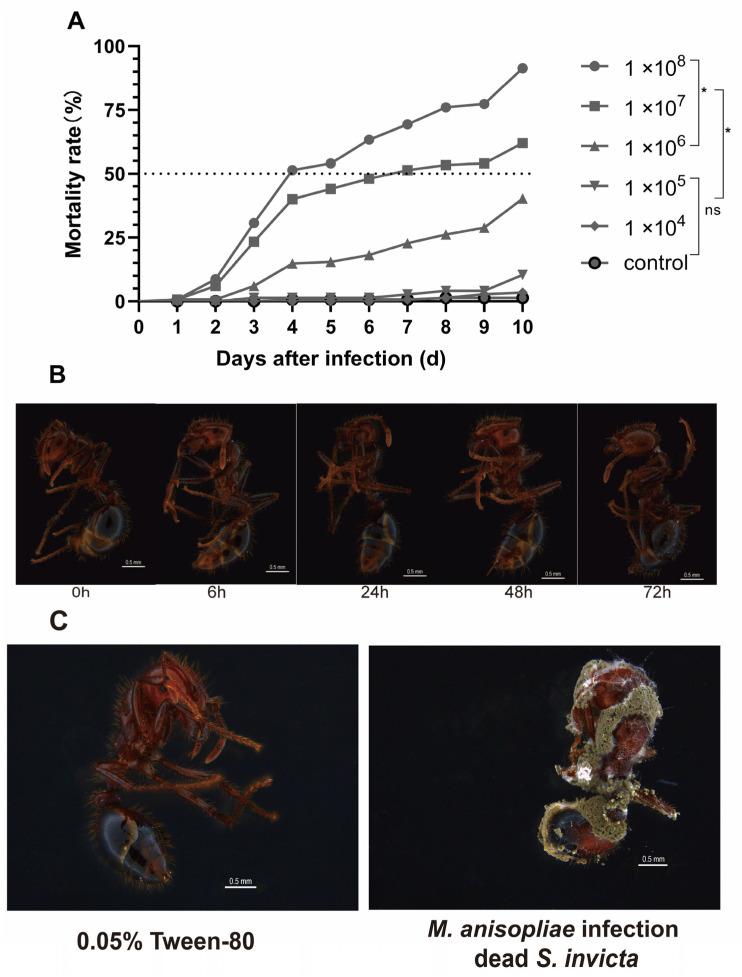
*M. anisopliae* induced mortality in major worker ants of *S. invicta*. (**A**) Mortality was recorded daily until ten days after exposure to different concentrations of *M. anisopliae*, while 0.05% Tween-80 was taken as the control. Dotted line represents a 50% mortality rate. “*” indicate significant differences at *p* < 0.05, and “ns” indicate insignificant differences. (**B**) Schematic representation of the progression of *M. anisopliae* infection in major workers of *S. invicta* at different time intervals (0 to 72 h). (**C**) The corpses of *S. invicta* infected with *M. anisopliae* were observed 96 h post-mortem. The left image shows an uninfected dead *S. invicta*, while the right image depicts a dead *S. invicta* infected with *M. anisopliae*.

**Figure 2 insects-14-00701-f002:**
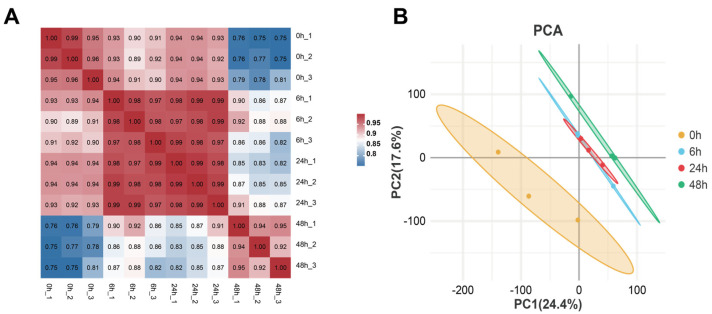
Global analysis of transcriptome data. (**A**) Heatmap of duplicate samples of different comparing groups of *S. invicta*. The color spectrum, from blue to red, represents Pearson correlation coefficients ranging from 0.75 to 1, indicating low to high correlations. (**B**) Principal component analysis (PCA) of the transcriptome for different comparison groups of *S. invicta*, represented by different colors.

**Figure 3 insects-14-00701-f003:**
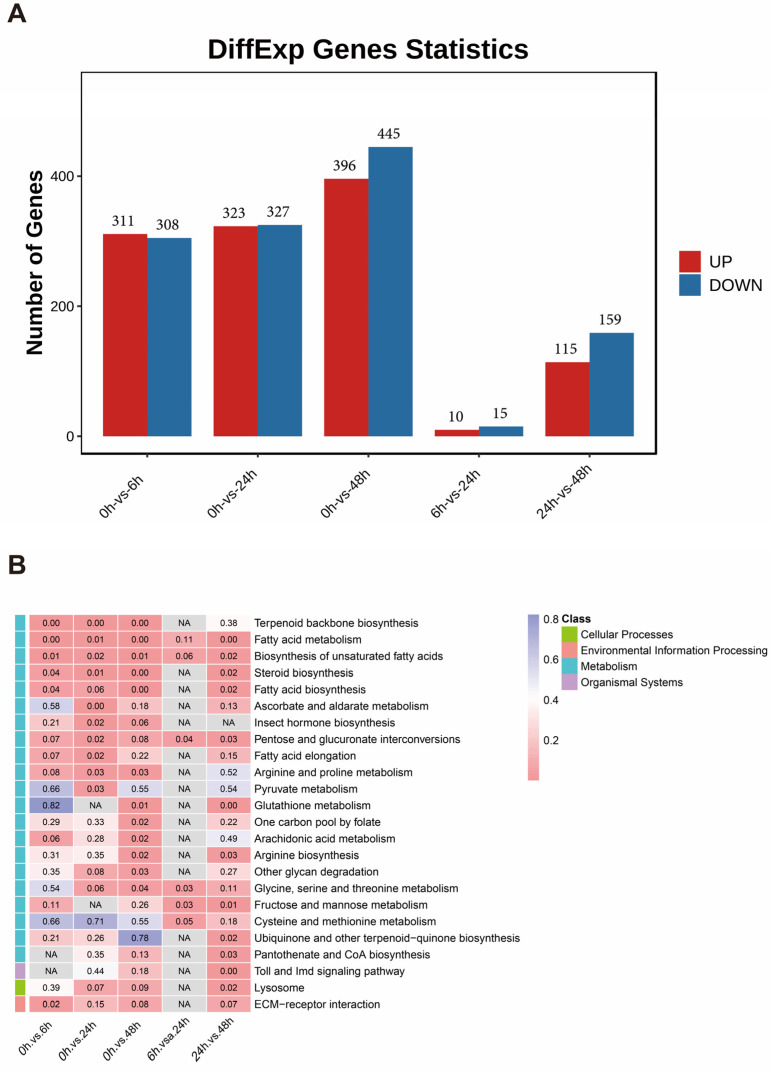
Statistics and KEGG analysis of DEGs between the control (0 h) and treatment groups (6 h, 24 h, and 48 h). (**A**) Bar plot representing the number of upregulated (red) and downregulated (blue) DEGs amongst different groups. (**B**) KEGG pathway enrichment analysis of genes from significantly expressed profiles. Each row corresponds to a pathway, each column corresponds to a comparing group, and each cell contains the *p*-value. The left color blocks indicate the first-level categories in the KEGG database. NA indicates that no genes were enriched in this pathway. The heatmap legend represents corresponding KEGG classifications.

**Figure 4 insects-14-00701-f004:**
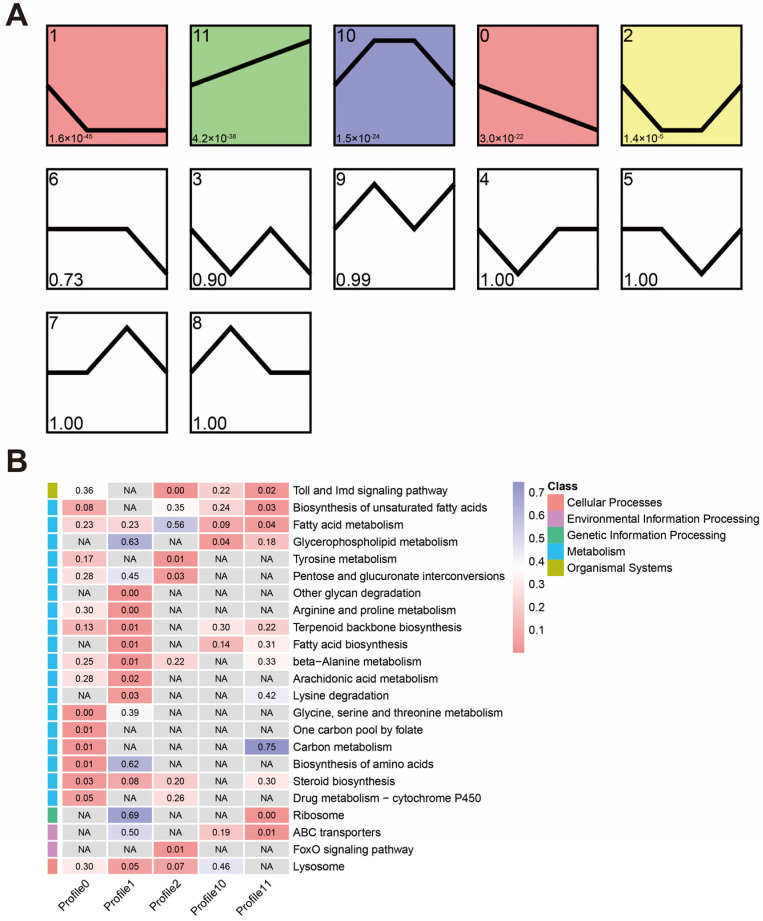
(**A**) Temporal expression trends of genes in *S. invicta* after *M. anisopliae* infection revealed by STEM analysis. Each box indicates a model profile, and the colored boxes represent significant profiles. The numbers in the box (top left) provide the order of the profile, and the *p*-value indicates significance (lower left). (**B**) KEGG pathway enrichment analysis of genes from significantly expressed profiles. Each row corresponds to a pathway, each column corresponds to a profile, and each cell contains the *p*-value. The left color blocks indicate the first-level categories in the KEGG database. NA indicates that no genes were enriched in this pathway. The heatmap legend represents corresponding KEGG classifications.

**Figure 5 insects-14-00701-f005:**
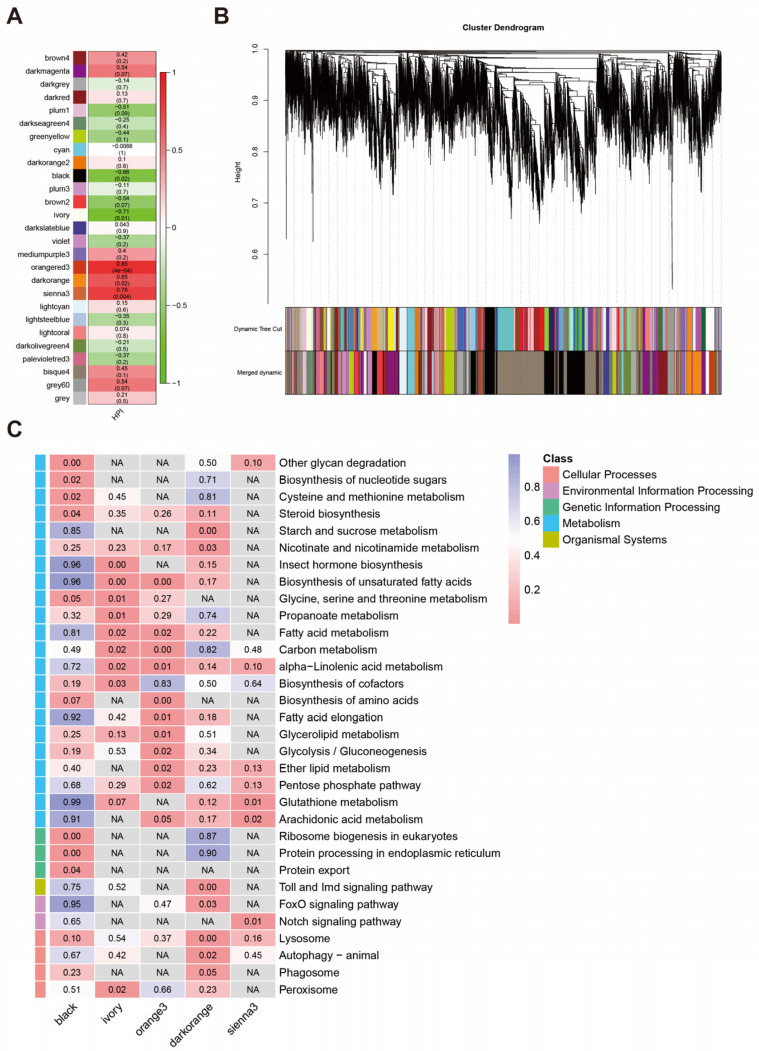
(**A**) Module–HPI associations. Each row corresponds to a module eigengene; each cell contains the corresponding correlation and *p*-value. (**B**) Hierarchical cluster tree showing the co-expression modules. (**C**) KEGG pathway enrichment analysis of genes from the significant model. Each row corresponds to a pathway, each column corresponds to a profile, and each cell contains the *p*-value. The left color blocks indicate the first-level categories in the KEGG database. NA indicates that no genes were enriched in this pathway. The heatmap legend represents corresponding KEGG classifications.

**Figure 6 insects-14-00701-f006:**
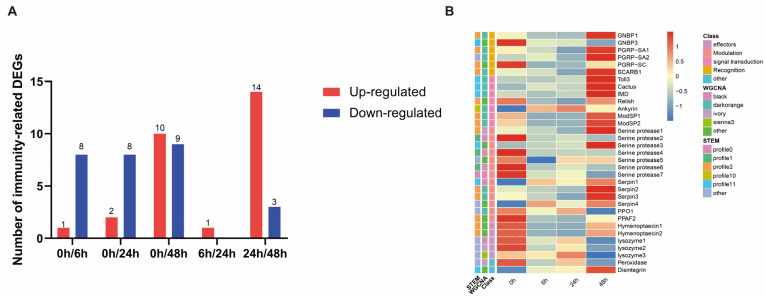
Immunity-related genes at different time intervals in *S. invicta* after *M. anisopliae* infection. (**A**) Bar plot represents the number of upregulated (red) and downregulated (blue) genes (**B**) Heatmap depicting the time course expressional profile of several immune-related genes. The heatmap legend corresponds to the KEGG classifications, the expression profiles, and the WGCNA modules.

**Figure 7 insects-14-00701-f007:**
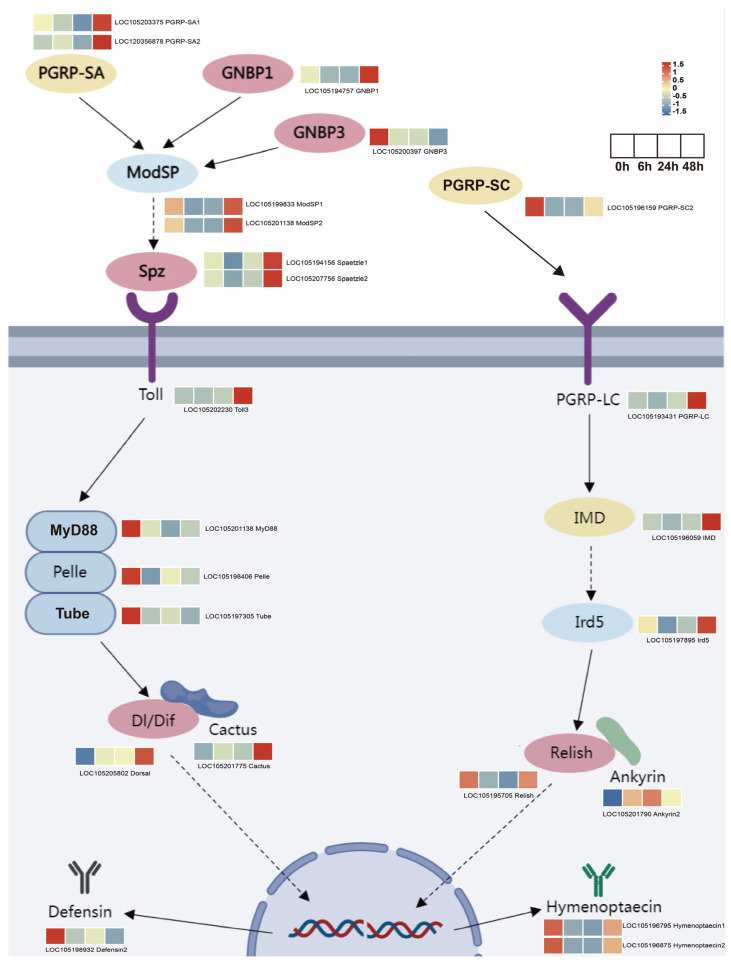
Pathway map displaying the gene expression patterns of key genes involved in the Toll and Imd signaling pathways. The color scale ranging from blue to red (row min to row max) indicates the expression of genes, presented as FPKM and normalized by Log2.

## Data Availability

A total of 12 RNA-seq raw reads from this study have been uploaded to NCBI with accession number PRJNA981300.

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
