# Peer review of "Transcriptomic Analysis Reveals the Impact of the Biopesticide *Metarhizium anisopliae* on the Immune System of Major Workers in *Solenopsis invicta"

_insects, 2023, doi:10.3390/insects14080701_

Round 1

Reviewer 1 Report

Please find my comments below:

Line 12: The binomial name should be followed by the author's name and year.

Line 31: Use either the scientific name or common name after introducing both at the beginning. Check throughout the paper for consistency.

Line 37: The authors mentioned billions of dollars in losses due to RIFA. Incorporate statistics, such as the value of losses and the cost of management, in the first paragraph of the introduction.

Line 45: More explanation is needed. For example: "The use of chemical insecticides, such as [specific chemical names], can cause harm in the following ways. Therefore..."

Line 56: Include references if there are any cases of resistance development in the use of bio-pesticides to control RIFA.

Line 66: The scientific name should be italicized. Check for consistency throughout.

Line 68-71: The last couple of sentences are more of results. Instead, introduce your hypotheses, potential outcomes, and assumptions.

Line 81-82: Include references if available.

Line 86-91: Did you follow a specific protocol or references to develop this protocol? If so, provide the references. Although it is common practice, it's important to provide any relevant references you might have followed.

Line 93: What was the basis for selecting these five concentrations? Explain why this rate was chosen. Were there any preliminary studies?

Line 104: Why was the rate of 5*10^7 spores/ml selected for RNA extraction and sequencing? Although it may be a result from the initial study of infection (method 2.2) and the LC50 value, the general audience might have difficulties understanding it.

Line 178-179: Briefly explain the reason for choosing the rate (for RNA-seq experiments) in the method section, as mentioned in the previous comment.

Line 180-184: It seems like methods were included here in the results section.

Figures: Ensure that scientific figures are accessible to readers with color-blindness. If possible, make them more accessible. Contact your journal about their requirements.

Line 195-350: Make sure to place methods in the methods section, not in the results section.

Line 229-235: "In the comparison between 24 h and 48 h... combat pathogens." Include the results in the results section. For example, if you found that the lysosome pathway was enriched in the 24h vs 48h comparison, state that in the results section. Discuss the association and involvement of different immune systems in the discussion section. Clarify the sections so that readers know what to look for in specific sections. Avoid sentences that start with "it suggests, it signifies, this indicates, our results suggest that," or any conclusions derived from results. Include them in the discussion section.

Line 371-390: The paragraph seems to focus more on what was done and what was found. However, it should emphasize what was found and provide possible explanations for the immune system or pathways. Therefore, state suspicions or confirmations about the defense mechanism in RIFA against fungus. Rewrite it to make more sense. Same goes with other paragraphs.

Line 467-475: What recommendations can be made for future research? Is there a need for specific follow-up research? If the potential of this research is so significant, why is it not yet integrated into practice? Incorporate these points.

Author Response

Below is a detailed response addressing the reviewers' comments and concerns.

Reviewer: 1

Comment 1: Line 12: The binomial name should be followed by the author's name and year.

Author response: Thank you for your suggestion. We have added the author's name and year to the binomial name in line 12 and included the author's name and year after each first mention of a binomial name throughout the article. (Line 13, 15, 52, 71-76, 79, 102, 425, 503, 520)

Comment 2: Line 31: Use either the scientific name or common name after introducing both at the beginning. Check throughout the paper for consistency.

Author response: Thank you for your kind reminder. We have thoroughly proofread the whole article and consistently made revisions to use scientific names.

Comment 3: Line 37: The authors mentioned billions of dollars in losses due to RIFA. Incorporate statistics, such as the value of losses and the cost of management, in the first paragraph of the introduction.

Author response: We have added in the first paragraph of the introduction the rapid expansion of RIFA in China and the overall economic losses caused by them (Lines 56-58). We have also included a citation (Reference 5) to support these statistics.

Comment 4: Line 45: More explanation is needed. For example: "The use of chemical insecticides, such as [specific chemical names], can cause harm in the following ways. Therefore..."

Author response: Thank you for your suggestion. We have made the necessary additions in the introduction (Lines 63-67) to emphasize the potential health hazards of certain pesticides on human beings. Additionally, we have included two references (References 11,12) to substantiate this claim.

Comment 5: Line 56: Include references if there are any cases of resistance development in the use of bio-pesticides to control RIFA.

Author response: Thank you for your suggestion. Up to this point, we have been unable to locate a reference specifically addressing resistance development in RIFA against bio-pesticides. However, evidence indicates effective immune responses against certain biocontrol agents. We have updated the information with examples from an immunological perspective (Lines 76-80) and have included supporting references (References 25,26).

Comment 6: Line 66: The scientific name should be italicized. Check for consistency throughout.

Author response: Thank you for your kind reminder. We have proofread the entire text and made corrections to such errors. The corresponding revisions have been highlighted in red.

Comment 7: Line 68-71: The last couple of sentences are more of results. Instead, introduce your hypotheses, potential outcomes, and assumptions.

Author response: Thanks for your valuable suggestion. We have revised this paragraph to emphasize our potential interpretations of the research findings and their practical implications. (Line 92-96)

Comment 8: Line 81-82: Include references if available.

Author response: Yes, here we adopted the criteria for Major worker ants based on a previous study, and we have provided the corresponding references in our article. (Reference 31) (Line 107).

Comment 9: Line 86-91: Did you follow a specific protocol or references to develop this protocol? If so, provide the references. Although it is common practice, it's important to provide any relevant references you might have followed.

Author response: As you mentioned, we referred to previous studies when formulating this protocol. We have included the references in our article. (Reference 33) (Line 116).

Comment 10: Line 93: What was the basis for selecting these five concentrations? Explain why this rate was chosen. Were there any preliminary studies?

Author response: Yes, as you mentioned, our lab has been extensively involved in studying M. anisopliae in recent years, utilizing it to infect various other pests, such as Plutella xylostella, Aedes albopictus, etc. Based on our experience with the fungi and several pilot bioassays conducted on S. invicta, we established an effective concentration range (104-1010). Additionally, we detected the mRNA transcript of immune genes by RT-qPCR and observed the mycelial growth on the cuticular surface of dead S. invicta at different time intervals and concentrations. By combining all these preliminary findings, we could establish time points and concentrations.

Comment 11: Line 104: Why was the rate of 5*10^7 spores/ml selected for RNA extraction and sequencing? Although it may be a result from the initial study of infection (method 2.2) and the LC50 value, the general audience might have difficulties understanding it.

Author response: Thank you for your kind reminder. As described in response to comment 10, we have added detailed and clear information (Lines 139-145).

Comment 12: Line 178-179: Briefly explain the reason for choosing the rate (for RNA-seq experiments) in the method section, as mentioned in the previous comment.

Author response: Thank you for your kindly remind. We have added the method in the RNA-seq experiments. (Line 139-145)

Comment 13: Line 180-184: It seems like methods were included here in the results section.

Author response: Thank you for your valuable suggestions. We have incorporated this description into section 2.2 of the methodology section. (Line 131-137)

Comment 14: Figures: Ensure that scientific figures are accessible to readers with color-blindness. If possible, make them more accessible. Contact your journal about their requirements.

Author response:  Thank you for your valuable input and suggestion. We have updated the color schemes of the figures (Figure1A and Figure S1), although for some figures it is essential to keep the various colors for clearer explanations. It is kindly requested to allow them in their current form. Thank you.

Comment 15: Line 195-350: Make sure to place methods in the methods section, not in the results section.

Author response: Thank you very much for your quality suggestion. We have proofread the entire manuscript and moved sentences from the result section to methodology. The corresponding modifications have been highlighted in red. (Line 178-181, 332-334)

Comment 16: Line 229-235: "In the comparison between 24 h and 48 h... combat pathogens." Include the results in the results section. For example, if you found that the lysosome pathway was enriched in the 24h vs 48h comparison, state that in the results section. Discuss the association and involvement of different immune systems in the discussion section. Clarify the sections so that readers know what to look for in specific sections. Avoid sentences that start with "it suggests, it signifies, this indicates, our results suggest that," or any conclusions derived from results. Include them in the discussion section.

Author response: Thank you very much for your suggestions. We have carefully revised the Results section, moved the interpretation of the results to the Discussion section, and improved the quality with additional information and references (3.3: Line 270-271, 403-409) (3.4: Line 286-292, 413-428) (3.5: Line 313-320, 428-443).

Comment 17: Line 371-390: The paragraph seems to focus more on what was done and what was found. However, it should emphasize what was found and provide possible explanations for the immune system or pathways. Therefore, state suspicions or confirmations about the defense mechanism in RIFA against fungus. Rewrite it to make more sense. Same goes with other paragraphs.

Author response: Thank you for your feedback on our draft. We have revised the paragraph to focus more on explaining the results of STEM and WGCNA from the perspective of host immunity. In the following paragraphs (Line 413-443), we have also added a discussion on the consequences and potential explanations of specific gene changes in terms of immune function (Line 444-447, 470-484, 597-501, 526-529).

Comment 18: Line 467-475: What recommendations can be made for future research? Is there a need for specific follow-up research? If the potential of this research is so significant, why is it not yet integrated into practice? Incorporate these points.

Author response: Thank you for your good suggestion. M. anisopliae has gained widespread adoption as an eco-friendly biopesticide for pest control. In this study, we employed RNA-seq to investigate the effects of M. anisopliae on the immune systems of S. invicta at different time points. Our research findings demonstrated that M. anisopliae suppressed the immune response of S. invicta during the early stages (6 and 24h) while stimulating its immune response at later stages (48 h). In ongoing and future research, we aim to confirm the function of these immune-related genes by RNAi or miRNA, or lncRNA.

Moreover, social immunity plays a critical role in social insects, and S. invicta employs various behaviors to prevent the spread of M. anisopliae within ant colonies. These behaviors, including self-grooming, grooming of nestmates, and carrying dead individuals' bodies, pose challenges to the large-scale application of M. anisopliae for controlling S. invicta. To gain a more comprehensive understanding of the mechanisms behind S. invicta's defense against M. anisopliae, we will conduct further in-depth data mining on RNA-seq data to identify genes associated with social immunity in S. invicta.

To address the abovementioned issues, we propose considering M. anisopliae as a component in pesticide formulations rather than using it separately. Alternatively, we can explore genetic engineering to construct recombinant fungi or combine M. anisopliae with other microbial insecticides, such as viruses or Bt, to create a highly virulent and fast-acting biopesticide. These approaches hold promising potential for enhancing the efficacy and sustainability of pest control strategies using M. anisopliae. (Line 542-553)

Reviewer 2 Report

L66: Italicise S. invicta

L67: Also italicise M. anisopliae. Across the manuscript, many scientific names have not been italicised; therefore, please check properly and correct them if they are required to dos so.

L87: Not clear with the sentence. Is the culture established from original monoconidium?

L92: Need detail- how did you prepare the fungal conidial concentrations? Is it in water or Tween 80 solution, if so, what is the concentration of Tween 80 (need registered name) in sterile water or etc?

L101: Need detail of how the data were collected, any data correction needed, is it analysed via ANOVA or regression or correlation or Post hoc test etc.

 L170: The title is not consistent to that in methodology section. Since the entomopathogenic fungus that is considered in this study is M. anisopliae, which you have already introduced, please directly mention M. anisopliae, not entomopathogenic fungus henceforth.

L188: Please italicise the scientific names, such as S. invicta or M. anisopliae.

L189: Not proper writing on the caption, for example aqueous tween, which should be scientifically represented by giving registered name of Tween 80 etc,

L191: this is not a schematic diagram, yet these are real images. For the visualised purpose, these juxtaposed images do not look different for me. If you could put a sporulated cadaver, that would be relevant.

L352: Should start an opening sentence that is directly linked to the result finding.

Conclusion:

This study covers significant experiments, particularly focusing on host insect genetic defence mechanism conferred against insect pathogen, e.g., M. anisopliae. However, the improvements on grammar and sentence structure are advised and some of lacunae to be addressed are highlighted in above.

English is good.

Author Response

Below is a detailed response addressing the reviewers' comments and concerns.

Reviewer: 2

Comment 1: L66: Italicise S. invicta

Author response: Thank you for your kind reminder. We have proofread the entire manuscript and corrected the errors where scientific names were not italicized.

Comment 2: L67: Also italicise M. anisopliae. Across the manuscript, many scientific names have not been italicised; therefore, please check properly and correct them if they are required to dos so.

Author response: Thank you for your valuable suggestion. We have reviewed the entire article and made revisions.

Comment 3: L87: Not clear with the sentence. Is the culture established from original monoconidium?

Author response: Yes, the subsequent culture was established from the original culture plates.

Comment 4: L92: Need detail- how did you prepare the fungal conidial concentrations? Is it in water or Tween 80 solution, if so, what is the concentration of Tween 80 (need registered name) in sterile water or etc?

Author response: Fungal conidial concentrations were prepared in 0.05% (v/v) Tween 80 (CMC = 0.012 mM (Sigma Aldrich P1754)) solution. A detailed methodology has been added to the manuscript. (Line 114-116)

Comment 5: L101: Need detail of how the data were collected, any data correction needed, is it analysed via ANOVA or regression or correlation or Post hoc test etc.

Author response: Thank you for your good suggestion. We have added the detail of how the data were collected (Line117-120). The mortality was corrected using Abbott’s formula. Five concentrations and control were compared via a log-rank test. The information has been updated in Lines 125-130.

Comment 6: L170: The title is not consistent to that in methodology section. Since the entomopathogenic fungus that is considered in this study is M. anisopliae, which you have already introduced, please directly mention M. anisopliae, not entomopathogenic fungus henceforth.

Author response: Thank you for your valuable input. We have revised the title to 'M. anisopliae infection in S. invicta', which is consistent with the methodology section. (Line 212)

Comment 7: L188: Please italicise the scientific names, such as S. invicta or M. anisopliae.

Author response: Thank you for your careful review. We have made corrections to the errors in the entire manuscript.

Comment 8: L189: Not proper writing on the caption, for example aqueous tween, which should be scientifically represented by giving registered name of Tween 80 etc,

Author response: Thank you for your kindly remind. We have replaced the aqueous tween with Tween-80. (Line 230)

Comment 9: L191: this is not a schematic diagram, yet these are real images. For the visualised purpose, these juxtaposed images do not look different for me. If you could put a sporulated cadaver, that would be relevant.

Author response: Thank you for your good suggestion. We have added Figure 1C, which includes a photograph of normal deceased S. invicta bodies alongside a photograph of dead S. invicta with spores, and provided corresponding descriptions and experimental methods. (Line 131-137, 221-226, 232-234)

Comment 10: L352: Should start an opening sentence that is directly linked to the result finding.

Author response: Thank you for your suggestion. We have now added a description at the beginning of the Discussion section that directly relates to our findings. (Line 380-390)

Round 2

Reviewer 1 Report

Goo job on revising the mansuscript nicely.